# Codon Bias Can Determine Sorting of a Potassium Channel Protein

**DOI:** 10.3390/cells10051128

**Published:** 2021-05-07

**Authors:** Anja J. Engel, Marina Kithil, Markus Langhans, Oliver Rauh, Matea Cartolano, James L. Van Etten, Anna Moroni, Gerhard Thiel

**Affiliations:** 1Membrane Biophysics, Department of Biology, Technische Universität Darmstadt, 64287 Darmstadt, Germany; anja.jeannine.engel@gmx.de (A.J.E.); mekithil@aol.com (M.K.); langhans@bio.tu-darmstadt.de (M.L.); Rauh@bio.tu-darmstadt.de (O.R.); matea.c@hotmail.de (M.C.); 2Nebraska Center for Virology, Department of Plant Pathology, University of Nebraska-Lincoln, Lincoln, NE 68583, USA; jvanetten1@unl.edu; 3Department of Biosciences, University of Milan, 20133 Milan, Italy; Anna.moroni@unimi.it

**Keywords:** codon usage, effect of synonymous codon exchange, membrane protein sorting, dual sorting

## Abstract

Due to the redundancy of the genetic code most amino acids are encoded by multiple synonymous codons. It has been proposed that a biased frequency of synonymous codons can affect the function of proteins by modulating distinct steps in transcription, translation and folding. Here, we use two similar prototype K^+^ channels as model systems to examine whether codon choice has an impact on protein sorting. By monitoring transient expression of GFP-tagged channels in mammalian cells, we find that one of the two channels is sorted in a codon and cell cycle-dependent manner either to mitochondria or the secretory pathway. The data establish that a gene with either rare or frequent codons serves, together with a cell-state-dependent decoding mechanism, as a secondary code for sorting intracellular membrane proteins.

## 1. Introduction

Each amino acid in a protein is on average encoded by about three synonymous codons. This provides a quasi-infinite sequence space of mRNA molecules and the potential of transmitting much more information than required for only coding the primary amino acid sequence. In this context it is well established that synonymous codons are used with distinct frequencies in different genomes [1] and that mRNAs encoding the same polypeptide with a codon bias can dramatically alter the amount of protein expression [2] including membrane proteins [3]. This phenomenon is already successfully used in biotechnology to increase protein production; similar codon-optimization strategies have also been proposed as therapeutic tools for tuning the cellular production of recombinant protein drugs, in mRNA therapies as well as in the production of DNA/RNA vaccines [4,5]. Such codon optimization strategies in medical therapy are however confounded by the fact that synonymous codons cannot be exchanged in all cases without affecting protein structure and function. There is increasing experimental evidence for a much more complex role of codon choice in that synonymous codons, for example, can alter mRNA splicing as well as mRNA folding and stability [6,7]. Codon choice is also known to regulate, together with the abundancy of the corresponding tRNAs, the velocity of protein synthesis, which can affect the proper folding of a nascent protein [4,8,9,10,11,12]. There are also isolated reports in which codon changes resulted in altered functional properties of proteins [13,14] and in some cases synonymous mutations have even been linked to diseases [4].

For a safe in vivo application of codon optimization strategies, it is important to better understand the potential impacts of codon choice on the cellular function of proteins. In this respect there is currently little information on the influence of codon usage on protein sorting in cells. Such an impact is not unrealistic considering that a biased frequency of synonymous codons can affect translation kinetics and co-translational protein folding and that both of these parameters can in turn alter protein sorting [15,16].

A good system for studying protein sorting of membrane proteins is two structurally similar algal viral-encoded K^+^ channels that have the structural hallmarks of eukaryotic K^+^ channel pores [17]. Consequently, they can utilize the protein sorting machinery of mammalian cells and hence they are able to provide an unbiased insight into this process independent of cellular coevolution. In this context, it was interesting that one of these channels, Kcv, is co-translationally sorted at the translocon into the ER [18], where it reaches the plasma membrane via the secretory pathway [19]. The second channel, Kesv, is sorted in a typical post-translational manner. It reaches its destination in the inner membrane of the mitochondria via the canonical TIM/TOM translocases without a mitochondrial targeting motif [20,21]. The decision between these two distinct trafficking pathways is made by the level of affinity of the nascent proteins for the signal recognition particle (SRP). While Kcv has a high binding affinity for the SRP, the other channel, Kesv, does not [22].

Additional studies have shown that sorting of the Kesv channel could be redirected by mutations in the second transmembrane domain of the channel; that is, the protein was no longer directed to the mitochondria but to the secretory pathway [20]. This redirection of mutated Kesv proteins occurs because the proteins become a substrate for the guided entry of the tail-anchored protein (GET) sorting pathway [22]. However, extensive mutational studies in the Kesv second transmembrane domain were not able to definitely identify an amino acid motif that was responsible for the difference in affinity for the GET factors [23].

To test the impact of codon choices on the sorting of the two virus-encoded K^+^ channels, we synthesized genes, which were codon-optimized for mammalian cells. For this purpose, the guanosine–cytosine (GC) content was increased and the majority of infrequently used codons were replaced by synonymous frequently used ones. The data show that co-translational sorting of the Kcv channel was insensitive to codon bias in the gene. The second channel Kesv, however, was sorted in mammalian cells in a codon-sensitive manner either to the mitochondria, the secretory pathway or even to both destinations in the same cell. The data support the hypothesis that codon bias in a gene can serve in combination with the primary amino acid sequence and with other cellular factors as a secondary code for sorting membrane proteins into one or the other pathway.

## 2. Materials and Methods

### 2.1. Codon-Modified DNA Variants of Channels

All codon-optimized DNA variants of Kesv and Kcv were obtained from GeneArt^®^ gene synthesis (ThermoFisher Scientific^TM^, Waltham, MA, USA). The DNA sequence of the randomized Kesv gene (Kesv_ran_) was generated using a Matlab script that randomly assigns to each amino acid of any primary sequence one of the codons coding for that amino acid from the corresponding group of redundant codons. As sorting of small membrane proteins is very sensitive to the structure of the nascent N-terminal polypeptide chain, which emerges from the ribosome [14] and since an N-terminal GFP tag is more likely to corrupt protein sorting than a C-terminal tag [24], all channel variants were exclusively tagged on the C-terminus via a linker to eGFP and inserted in a peGFP-N2 vector. For optogenetic studies the channels were inserted in a pGL4-C120-Fluc vector (as described in [25]) in exchange for the firefly luciferase. Codon sequences of the channel constructs, linkers and eGFP are shown in Appendix A. All experiments were performed with linker 1 (Figures 1–5 and Figure 6A), linker 2 was used for data in Figure 6B,C.

### 2.2. Mutagenesis

To create channel chimeras, a chimeric PCR was performed [26]. For chimeras in which entire DNA sequence segments were exchanged, two or three desired gene fragments were initially amplified from the corresponding DNA templates. In order to ensure the fusion of the gene fragments, overhangs were created over the primers, which were complementary to the adjacent gene fragment. Interfaces for restriction enzymes were introduced via the overhangs of the outermost fragments in order to subsequently enable ligation with the vector. The gene fragments were then fused together with a second PCR. The Phusion DNA polymerase (ThermoFisher Scientific^TM^; Waltham, MA, USA) was used for all described PCR approaches according to manufacturer specifications. All PCR products were electrophoretically separated in a 1–2% agarose gel in 1× TAE (Tris, acetate, EDTA) and purified using the Zymoclean^TM^ Gel DNA recovery Kit (Zymo Research; Irvine, CA, USA) according to the manufacturer’s specifications. The DNA concentrations were photometrically determined using the Nano-Drop^®^ ND-1000 spectrometer (PeQlab Biotechnologie GmbH; Erlangen, Germany).

After fusion of gene fragments via PCR and subsequent purification, both the empty peGFP-N2 and the final PCR products were first treated with the respective Fast Digest^®^ restriction enzymes (ThermoFisher Scientific TM; Waltham, MA, USA) according to the manufacturer’s specifications. In the next step, the PCR product was ligated into the cleaved vector using T4 ligase (ThermoFisher Scientific^TM^; Waltham, MA, USA) according to the manufacturer’s specifications. The full ligation product was used for the transformation of competent *E. coli* DH5α cells by heat shock. Finally, the transformed *E. coli* were plated on LB kanamycin plates and incubated overnight at 37 °C.

The colonies were used to inoculate LB medium liquid cultures with 100 μg/mL kanamycin. On the following day, the plasmid DNA was purified using the ZR Plasmid Miniprep^TM^ Classic Kit (Zymo Research; Irvine, CA, USA) and sequenced (Eurofins MWG Operon GmbH Ebersberg, Germany). The sequencing was controlled using SnapGene software (GSL Biotech; Chicago, IL, USA).

### 2.3. Heterologous Expression

Localization of the eGFP tagged channels was performed in human embryonic kidney (HEK293) cells. In addition, CHO, HeLa, HaCaT and COS-7 cells were used. All cell lines were cultured at 37 °C and 5% CO_2_ in T25 cell culture flasks in an incubator with the appropriate culture media (see below). For imaging, cells were placed 48 h prior to examination on sterilized glass coverslips (No. 1.0; Karl Hecht GmbH & Co. KG, Sondheim, Germany) with a Ø = 25 mm. The cells were incubated for ~24 h at 37 °C with 5% CO_2_. As soon as the cells reached a confluence of 60%, they were transfected with the appropriate plasmids. GeneJuice (Novagen, EMD Millipore Corp.; Billerica, MA, USA) or TurboFect^TM^ (Life Technologies GmbH; Darmstadt, Germany) were used as transfection reagents according to manufacturer specifications. Unless otherwise stated, 1 μg of plasmid DNA of the corresponding construct was always used or, in the case of co-transfection, 0.5 μg of each of the desired constructs.

For experiments on temperature dependence, a media change with Leibovitz (1×) L-15 medium was performed 4 h post transfection after which cells were incubated overnight at a desired temperature without additional CO_2_. In all experiments where cells were incubated at different test temperatures, a control batch was treated in the same manner and incubated at 37 °C.

In experiments on the influence of the metabolic status on protein sorting, a change of medium to the respective test medium with a different content of sugars (4.5 g/L or 1 g/L glucose) was performed 4 h after transfection. Cells were then incubated overnight at 37 °C with 5% CO_2_. In each experiment on the influence of the metabolic status on protein sorting, a control in HEK293 standard medium was conducted with the same treatment.

For illumination of cells in which the channel protein was expressed under control of the light-sensitive EL222 system, a custom-made illumination setup was designed with six 450 nm LEDs (Winger WEPRB3-S1 Power LED Star, 3W, Nettetal, Germany) arranged on a plate to fit under a standard 6-well cell culture plate. For tuning illumination times, the LEDs were attached to a timer that allowed application of light pulses of defined length on a second to minute timescale. For the experiments shown here, pulsed blue light of 120 µE was applied for 16 h prior to imaging. Light pulses were either 10 s or 20 s long followed by 60 s of darkness.

### 2.4. Cell Culture Media

HEK293, COS-7 and HeLa: DMEM/F12-Medium with Glutamine (Biochrom AG, Berlin, Germany) plus 10% fetal calf serum (FCS) und 1% Penicillin/Streptomycin. HaCaT: DMEM-Medium with 4.5 g/L glucose plus 2 mM Glutamine (Biochrom AG, Berlin, Germany), 10% FCS and 1% Penicillin/Streptomycin.

### 2.5. Confocal Laser Scanning Microscopy (CLSM)

Initial microscopic screening and quantitative examination of protein sorting in cultured mammalian cell lines was performed on a confocal Leica TCS SP5 II microscope (Leica GmbH, Heidelberg, Germany). Unless stated otherwise, cells were kept with 500 μL PBS medium (8 g/L sodium chloride, 0.2 g/L potassium chloride, 1.42 g/L disodium hydrogen phosphate, 0.24 g/L potassium hydrogen phosphate; pH was adjusted with 1M sodium hydroxide up to 7.4) on coverslips, clamped into a custom-made aluminum cup at least 16 h after transfection.

Cells were imaged with a PL APO 100.0 × 1.40 oil immersion lens. Dyes or fluorescent proteins were excited with an argon laser (488 nm) or a helium-neon laser (561 nm) and the emitted light observed at the following wavelengths: GFP: 505 nm–535 nm, MitoTracker^®^ Red FM and mCherry: 590 nm–700 nm, ER-Tracker^TM^ Red (BODIPY^®^ TR Glibenclamide): 600 nm–700 nm.

The primary observation of the eGFP-tagged channel localization in cells was always carried out without fluorescent labeling of the target membranes. This should exclude any influence by the overexpression of a compartment specific protein. For detailed localization studies the mitochondria or the ER were labeled either with fluorescent dyes or organelle specific marker proteins. Dye labeling was performed according to established manufacturers’ protocols. The growth medium was replaced with PBS containing the organelle-specific dyes MitoTracker^®^ Red FM (25 nM) or ER-Tracker^TM^ Red (BODIPY^®^ TR Glibenclamide) (1µM) (Life Technologies GmbH, Frankfurt, Germany). After incubation with MitoTracker^®^ Red FM for 5 min or with ER-Tracker^TM^ Red (BODIPY^®^ TR Glibenclamide) for 10 min, cells were washed with fresh PBS incubation buffer before imaging. As organelle-specific dyes have only a limited specificity mitochondria and ER were in all experiments also labeled with fluorescent specific marker proteins. The subunit VIII of human cytochrome C oxidase fused with the fluorescent protein mCherry (COXVIII::mCherry) was employed to label the inner membrane of the mitochondria. The ER retention sequence HDEL fused with fluorescent protein mCherry (HDEL::mCherry) was used to label the ER. Both plasmids were obtained from Addgene (Cambridge, MA, USA); mCherry-Mito-7 and mCherry-ER-3 were kindly provided by Michael Davidson (Addgene plasmids #55102 and #55041). Labeling of the target organelles with fluorescent proteins and fluorescent dyes provided overall the same results on sorting of the channel proteins. For a quantitative analysis of channel sorting only the fluorescent dyes were used in order to avoid an interference with the sorting of the organelle marker protein and the channel protein of interest.

For a quantitative estimate of protein localization in different cell compartments, images of at least 100 individual cells with a fluorescent signal were recorded. The classification of protein sorting was done manually according to the criteria described in Figure 1. To limit the bias of manual classification most images were independently analyzed by at least two experimenters. The results of three experimenters on the relative distribution of Kesv_wt_ varied over 15 independent experiments with a total of > 600 cells examined by less than 2%. To further test reproducibility, the same images on the complex expression pattern of chimera 1 were in a blinded manner examined by three independent and trained individuals. The resulting values for the relative distributions varied in this case by ≤ 4%. Image analysis was generally performed using LAS AF Lite software (Leica Microsystems GmbH, Wetzlar, Germany) or Fiji [27]. Images were created using either IGOR Pro 6 (Wavemetrics, Tigard, OR, USA) or Origin 9 (OriginLab, Northhampton, MA, USA).

### 2.6. Cell Cycle Analysis

Cell cycle analysis was performed on fixed HEK293 cells transfected with Kesv_wt_::eGFP and Kesv_op_::eGFP (or Kesv_wt_ and Kesv_op_ without the eGFP tag) using propidium iodide staining. The cells were harvested >18 h after transfection by trypsination, transferred to 5 mL PBS and centrifuged for 6 min at room temperature (RT) at 1000 rpm. The cell pellet was re-suspended in 0.5 mL PBS. Cells were fixed in 4.5 mL 70% ethanol at −20 °C and under continuous vortexing. The fixed cells were centrifuged 5 min at 1000 rpm at RT, re-suspended in 5 mL PBS and again pelleted 5 min at 1000 rpm and RT. The cell pellet was then re-suspended in 1 mL staining solution (10 mL 0.1% Triton × 100 in PBS, 2 mg RNAse A, 200 μL 1 mg/mL Propidium Iodide (PI)) and incubated for 30 min at RT. The cells were pelleted again before measurement, resuspended in 0.5 mL PBS and filtered through a 40-μm filter. The measurements were performed using the blue laser (488 nm) of the S3e Cell Sorter (Bio-Rad GmbH; Munich, Germany). The cell cycle phases were determined with the FlowJo (FlowJo, LLC; Ashland, OR, USA) software by analyzing the PI height to PI area distribution.

### 2.7. Software Analysis

Almost all amino acids are encoded by more than one codon. The %MinMax algorithm valuates in a species-dependent manner the relative rareness of a nucleotide sequence, which codes for a protein of interest [28]. We used this algorithm to estimate the codon bias of the channel proteins including linker and eGFP in human cells. Data were calculated as a moving average over a window of 18 codons where 0% represents the least common and 100% the most common codon.

## 3. Results

### 3.1. Mitochondrial Sorting of Channel Protein Is Modulated by Codon Choice

To examine the influence of codon bias on Kesv sorting, we compared its location in HEK293 cells after expressing the GFP-tagged protein from a wild type (Kesv_wt_) gene, a gene that was codon-optimized for expression in mammalian cells (Kesv_op_) and a gene with a randomized sequence of favorable/unfavorable codons (Kesv_ran_) (Appendix A). Cells transfected with the wt gene can be grouped into three distinct populations: (i) a majority of cells with the GFP-tagged protein being targeted to the mitochondria in a background of GFP fluorescence in the cytosol (Figure 1A,C), (ii) a small number of cells with the channel in the secretory pathway (SP) (Appendix A and Figure 1C), and (iii) cells with a strong GFP signal throughout the cell (Appendix A, Figure 1C). The latter include GFP in the nucleus and exclude it from mitochondria and peri-nuclear ring (Appendix A). We interpret the presence of GFP in the nucleus as evidence for partial degradation of the channel/GFP construct; only a cleavage of GFP from the hydrophobic channel will allow diffusion of the fluorescent tag into the nucleus [29].

The robust pattern of sorting was altered by expression of the codon-optimized gene (Kesv_op_): The protein was no longer targeted to the secretory pathway but sorted with increased propensity to the mitochondria (Figure 1B,C). An enhanced tendency for mitochondrial sorting was further underscored by quantifying the relative fluorescence in mitochondria versus background fluorescence in the cytosol (Figure 1B,D). The latter ratio is about six times higher in cells transfected with the codon-optimized gene compared to those transfected with the wt gene.

In contrast, expression of randomized Kesv (Kesv_ran_) strongly lowered the probability for mitochondrial sorting below that of the wt gene (Figure 1C). This reduction occurred because of an increased frequency in protein degradation and increased sorting to the SP. Taken together, the data show that the same protein can be targeted to different cellular locations in HEK293 cells and that their sorting destiny depends both on the codon structure of the gene and on additional cellular factor(s). This translates the same message even in adjacent cells in one case into a protein sorting to the mitochondria and in the other case into a protein sorting to the SP (Appendix A).

We conducted several experiments to determine if the phenomenon of codon-sensitive sorting could be a technical artifact of the experimental system. A first set of control experiments shows that the presence of the mitochondrial marker has no impact on sorting (Figure 2A). A co-expression with the ER marker HDEL::mCherry in contrast favors mitochondrial sorting of the channel and eliminates targeting into the ER. The data suggest that a competition for co-translational sorting into the ER is influencing the targeting of the channel in our system (Figure 2A). For this reason, all quantitative data were obtained in the absence of protein markers. We further reasoned that, if the sorting phenomenon is dominated by an overload of the translation and/or sorting system, a further increase in DNA should mimic the effect of codon optimization in the wt channel. A 10-fold difference in DNA concentration used for transfection however had no appreciable impact on sorting of Kesv_wt_ and Kesv_op_ (Figure 2B). This suggesting that the difference in sorting between the two constructs is not dominated by an oversaturated sorting system. Next, we addressed the question if rate-limiting sorting factors like the signal recognition particle (SRP) are over engaged when expressing a codon-optimized gene and that this could alter sorting. We therefore transfected HEK293 cells with either the wt gene or a codon-optimized gene of a second small K^+^ channel from chlorella virus PBCV1 (Kcv_PBCV1_) [30] (Appendix A). In this case codon optimization had no impact on sorting (Figure 2C), suggesting that the general targeting of a small channel protein is not corrupted by codon bias per se. That is, expression of a codon-optimized channel protein does not alter targeting of the nascent protein to the mitochondria because of over engaged SRPs.

All control experiments support the idea that sorting of the Kesv channel is affected by the codon structure of the Kesv gene and there is no evidence to suggest that this sorting phenomenon is a technical artifact of the experimental system. Based on current knowledge on the effects of synonymous codon usage, this could be related to several modes of action including codon-sensitive efficiency and stringency of mRNA decoding, the synthesis and stability of mRNA or the translation velocity and folding of nascent proteins [4,8,9,31].

### 3.2. Codon-Biased Sorting Is a General Phenomenon of Mammalian Cells

To test if codon-sensitive sorting of Kesv is peculiar to HEK293 cells, we repeated the experiments from Figure 1 with four other cell lines. These experiments were motivated by the fact that established mammalian cells lines differ among other features in their expression levels of proteins [32] as well as in the trafficking of heterologously expressed membrane proteins [33]. Relevant for the present investigation is also that different tissues exhibit distinct differences in their tRNA concentrations, a feature which could affect protein sorting [34]. Transfection of all four different mammalian cells with the wt gene resulted in diverse sorting of Kesv to the mitochondria. While the sorting to the mitochondria was strong in HeLa cells (Figure 3A), the channel was sorted to the mitochondria in only a few COS-7 and HaCaT cells (Figure 3B). In CHO cells the channel was not only found in the mitochondria but also in the secretory pathway (Figure 3C). The results of these experiments confirm the assumption that the efficiency of sorting to the mitochondria is not only due to the channel protein, but that it is also influenced by cellular factors.

The data in Figure 3D, however, clearly show that the effect of codon optimization on sorting to the mitochondria is conserved in all five mammalian cell lines. In all cell lines tested codon optimization created a strong tendency for sorting the channel to the mitochondria. This was independent of the cellular conditions, which are in different cell types more or less favorable for mitochondrial sorting of the wt channel.

### 3.3. Chimeras of Genes with Optimized/Non-Optimized Codons Cause Complex Sorting Patterns

We constructed chimeras consisting of codon-optimized and wt codons (Figure 4A) to identify a potentially critical region in the gene that is important for this phenomenon. Figure 4B shows the relative sorting of different chimeras in HEK293 cells. Remarkably chimeras of synonymous codons result in distinctly different sorting phenotypes. One important conclusion from the data in Figure 4B is that the protein is able to traffic in a codon-dependent manner either to the secretory pathway or to the mitochondria (Figure 4B). This pattern occurs in an inverse relationship in that the proteins are sorted to the mitochondria when they escape sorting to the secretory pathway. This suggests that the decision on sorting is the result of multiple competing factors.

A surprising observation is that the protein can occur in a codon-dependent manner (e.g.; Chimera C4) in the same cell in the SP and in the mitochondria (Figure 4B,C). The dual sorting of three chimeras (C-1, C-4, and C7) within one cell was confirmed by co-localization with the respective marker proteins (Figure 4C and Appendix A). The results of these experiments establish that the channel protein can be targeted to both the SP and to the mitochondria in the same cell and that this process is influenced by the codon structure of the gene. In this respect the chimera mimics the natural situation of channels like Kv1.3, which can occur in the plasma membrane and in the mitochondria [35].

Scrutiny of the data in Figure 4 did not reveal a single region in the gene in which codon optimality favors sorting to the mitochondria or to the SP. For example, optimization of the last 14 C-terminal codons (Chimera C6) has the same impact on sorting as optimization of the entire channel. Codon optimization of upstream regions (e.g., Chimera C3), however, can even be counterproductive and promote protein degradation and sorting to the SP. Additionally, optimization of a stretch of codons ≥ 30 codons from the start (Chimera C8), which might negatively affect binding of the nascent protein to the SRP when emerging from the ribosome [16], did not increase mitochondrial sorting.

The complex pattern in which a particular region can favor sorting to the mitochondria in one chimera but not in another suggests that the decision on sorting is the result of multiple competing factors. To test this assumption, we altered the sorting of Kesv in a codon bias-independent manner by inserting amino acids into the second transmembrane domain [20,22]. In a screening endeavor we inserted in position 113 of Kesv_wt_ 16 different triplets of randomly chosen amino acids and found that the triplet GML was the most potent in redirecting sorting of this mutant (Figure 4D). This channel (Kesv-113GML_wt_) was no longer detected in the mitochondria but was present in the SP in >80% of the cells (Figure 4D). When the full-length gene of Kesv-113GML was codon-optimized it had almost no effect on the sorting of the channel. Like the wt protein, Kesv-113GML_op_ was still efficiently sorted to the SP. The results of these experiments indicate that codon optimality does not per se favor sorting to the mitochondria. If a protein like Kcv or Kesv-113GML has a strong signal for trafficking to the SP, the sorting destiny is only slightly affected by codon optimality.

### 3.4. Impact of Codon Usage on Sorting

Based on current knowledge, the sorting of proteins between the secretory pathway and mitochondria is determined by fundamentally different mechanisms [36,37]. The canonical targeting mechanism involves interactions between the protein and specific sorting factors for subsequent delivery to membrane-embedded translocases [36,37]. For some short tail-anchored proteins, a spontaneous insertion in either the ER or the mitochondria membrane is known [38]. The latter pathway however only targets proteins to the outer membrane of mitochondria and is hence not relevant for the Kesv channel, which is sorted to the inner mitochondrial membrane [20]. A third targeting mechanism employs a distinct pre-sorting of the respective mRNA to the final destination where the protein is translated directly into its final location [36,39]. The complex sorting phenotypes of the different Kesv constructs could in principle result from an impact of codon usage on any of the three mechanisms. Codon optimization could alter the structure of a peptide signal or the interaction kinetics of the nascent protein with sorting factors. This could arise from codon sensitivity of translation efficiency, protein folding or transcript stability [7]. Alternatively, codon optimality could also alter the structure of the mRNA [40,41] and as a consequence perturb or create essential targeting codes for mRNA sorting [42].

To examine the impact of codon usage on the mRNA structure, we calculated the free energy of the different RNAs using an RNA structure prediction algorithm [43]. To augment the relevance of structural elements in the variable channel-coding region, the free energy was only calculated for this part of the construct, ignoring the contribution of the constant linker/GFP. A plot of the efficiency of sorting of the channel to the mitochondria as a function of this free energy implies that the codon choice has considerable impact on mRNA structure. This could generate altered targeting codes for mRNA sorting but could also affect translation velocity [39].

The linear relationship with a weak correlation (coefficient 0.44) between mRNA stability and channel sorting does not exclude a contribution of RNA stability to the differential sorting of the channel (Figure 5A). Still the data are not sufficient to explain sorting of different chimeras. The different chimeras have roughly the same free energy between −115 and −125 kcal/mol; one promotes sorting to the mitochondria in ca. 10% of the cells while the other chimera does it more than 70% of the time. The results of this analysis are in agreement with published data showing that differences in the secondary structure of mRNA are not sufficient to explain the causal link between codon bias and ribosome elongation rates [7].

To further test potentially hidden impacts of the mRNA structure on protein sorting, we expressed both channels under control of an optogenetic transcription system. It was reported that the transcription of a gene of interest could be triggered by light via a quasi-instant activation of a transcription factor [44]. Based on the kinetic relationship between transcription and subsequent translation [45,46], we reasoned that a light-triggered burst in transcription would neither affect the mRNA nor the structure of the translated protein but the kinetics of transcription/translation to such an extent that even the codon-optimized Kcv channel would be sorted to the mitochondria. To test this prediction, codon-optimized channels were expressed in HEK293 cells under control of the light-sensitive EL222 system. Hence, a light triggered burst in transcription should not affect protein sorting if the latter is determined by the structure of the RNA. To test if the same mRNA can generate in such a system differences in sorting, Kcv_op_ was expressed in HEK293 cells under control of the light-sensitive EL222 system. After triggering transcription by blue light, we monitored the distribution of the GFP-tagged channels in HEK293 cells. The data in Figure 5B show that the light-inducible system had a strong impact on the targeting of Kcv_op_. While the latter channel is preferentially sorted to the SP following conventional transfection (Figure 2), its sorting is shifted with an increased tendency for the mitochondria when transcription is triggered by light (Figure 5B). We are not able at this point to explain the mechanism, which is underlying the shift in sorting of Kcv_op_ in the context of the light-sensitive transcription system. However, the results of these experiments are in good agreement with the view that codon optimality is affecting one or more critical steps in the translation of the nascent channel proteins. They further confirm that differences in the secondary structure of the mRNA are unlikely crucial for sorting. The fact that the mRNA for the Kcv_op_ channel was the same for conventional expression or expression under control of the optogenetic system also indicates that codon choice has in the present system no impact on signals, which contribute to mRNA localization in cells [39].

In the case that codon optimality does not affect mRNA sorting but translation efficiency, folding or transcript stability [42], protein sorting should be sensitive to temperature; lower temperatures should slow translation efficiency and folding. To test this prediction, we followed the expression and sorting of Kesv_wt_ and the codon-optimized channel in HEK293 cells at 37 °C and 25 °C. The data in Figure 5C show that the incubation temperature indeed affects sorting of the channel. Cells transfected with the Kesv_wt_ and with the codon-optimized gene exhibited an altered sorting pattern at 25 °C compared to 37 °C. While the relative number of cells with the channel in the mitochondria decreased at 25 °C, the fraction of cells with a degraded protein increased. This result implies that sorting to the mitochondria is favored by higher temperature. The data are in agreement with the hypothesis that codon optimization alters cellular processes, which are also accelerated by temperature. Since a change in the incubation temperature is a rather non-selective parameter, which affects many cellular processes, the data do not yet provide a clear indication of the precise mechanism responsible for codon-sensitive sorting.

Since all experiments are based on transient transfection of channel proteins it is possible that the difference in their sorting simply reflects different amounts of proteins, which are generated in individual cells; this parameter might than be influenced by codon optimality. To test this possibility, we randomly chose 15 images like in Appendix A in which a cell with a mitochondria-sorted Kesv channel was in the same optical plane close to one with the channel sorter into the SP. Assuming that the fluorescence intensity (FI) of GFP is an indirect approximation of the amount of tagged protein [47], we defined outside of the nucleus as regions of interest (ROI) and measured their integrated fluorescence density (IFD). For each pair of cells, a ratio was calculated by dividing the IFD value from GFP in mitochondria by the respective value from GFP in the SP (IFD(ROI_MI)_/IFD(ROI_SP_)). The resulting ratios cover a wide range from 0.4 to 2.1, with a mean value of 1.2 ± 0.5 (Appendix A). The results of this analysis exhibit no apparent correlation between the amount of GFP-tagged protein in a cell and its destiny of sorting. This conclusion is further supported by a similar analysis of pairs of cells in which the channel was, in both cases, sorted to the mitochondria (Appendix A); in this case, the IFD values between adjacent cells deviate by a factor between one and four without an apparent consequence on sorting. Collectively, the data suggest that the protein concentration varies considerably between individual cells, but this has no systematic impact on sorting.

### 3.5. Sorting Is Affected by State of the Cell Cycle However, Not by the Energy Status of Cells

The experiments have so far shown that the K^+^ channel protein Kesv can be sorted in cells to two distinct destinations in a codon-dependent manner. However, this message is interpreted in different ways by individual cells. That is, in the same experiment one cell decodes this message as a signal for sorting to the mitochondria, whereas another cell sorts the channel with the same message to the SP. A similar scenario was previously reported for the Slit3 protein in which sorting to the plasma membrane or the mitochondria were determined by some state of cell differentiation or development [48].

One aspect in which cells in a non-synchronized culture differ is their position in the cell cycle. It has been shown that different states of the cell cycle correlate with different metabolic activities and different concentrations of tRNAs [49]. This led to the concept that availability of ribosomes, aminoacyl-tRNA synthetases and charged tRNAs might be rate limiting and influence the speed of protein translation in one condition but not in another.

To test this possibility, we treated HEK293 cells with RO-3306, a CDK1-inhibitor, which arrests cells in the G2 state [50]. The results from a flow cytometry analysis show that treatment with 7 µM RO-3306 caused the expected arrest of cells in G2 after 48 h of incubation (Figure 6A). While in control cells 36.3 ± 3.2% of the cells were in G2, 73.5 ± 1.5% of the cells were in G2 after inhibitor treatment. We then used cells, which were treated with the same protocol, to examine the sorting pattern of the Chimera C1 (Figure 6A,B). This Chimera was chosen because of its complex sorting pattern under control conditions (Figure 4B). We reasoned that any alteration in the cells would exhibit a strong impact on the sorting of this construct. An analysis of the sorting pattern from three independent experiments shows that the state of the cell cycle strongly influenced the fate of protein sorting (Figure 6A,B). An increase in cells in G2 goes together with a decrease in mitochondrial sorting. At the same time, this condition favors sorting of the channel to the SP. This inverse behavior of sorting to the mitochondria and SP is the same as in Figure 4B and underscores a causal relationship between both sorting pathways.

In an additional assay, we also incubated HEK293 cells with different concentrations of glucose in the medium. This has an impact on many cellular parameters including energy status [51], signaling pathways [52] and even the expression of membrane proteins [53]. When Chimera C1 was expressed in cells incubated in low, normal or elevated glucose concentrations, the channel exhibited overall the same complex sorting pattern (Figure 6C) as in Figure 4. Thus, addition or deprivation of glucose in the incubation medium of the cells, which presumably influenced among other factors the energy status of the cells, had no appreciable impact on sorting of the Kesv channel.

## 4. Discussion

Our analysis of the impact of synonymous codon choices on the sorting of two similar K^+^ channel proteins shows that codon bias has, in combination with cellular factors, a strong impact on the targeting destiny of one channel, Kesv, but not on the other, Kcv. A key message from these results is that the codon choice of the Kesv gene can serve in this orthogonal system as a signal for intracellular protein sorting. Hence, a gene sequence can contain more information for protein sorting than is encoded in the primary amino acid sequence. All of these data can be explained in the context of the redundancy in the genetic code in which most amino acids are coded by multiple synonymous codons.

Current knowledge on the role of synonymous codons provides several possible explanations on how they could affect protein sorting. This could occur at the level of stability or localization of mRNA, codon-sensitive efficiency and stringency of mRNA decoding, the synthesis, as well as the translation velocity and folding of nascent proteins [7,8,42]. At this point it is not possible to pinpoint which of these mechanisms is responsible for the sorting phenomena. However, from circumstantial evidence we reason that translational missense errors can be excluded as an explanation. It is known that synonymous codons exhibit different frequencies of translational misreading. While this is a frequent phenomenon in bacteria, it is very rare in eukaryotes [54]. The diversity of sorting phenomena obtained with the chimeras is not compatible with such a rare event. Our data are also not in agreement with an effect of codon usage on the stability of mRNA structures. A plot of sorting efficiency as a function of the estimated free energy of mRNA stability exhibits only a weak correlation. This suggests that this parameter could contribute to but not completely explain the different sorting patterns. Finally, the results provide no evidence for an impact on codon optimality on targeting signals in the mRNA, which could be responsible for the differential sorting. When the Kcv_op_ channel is expressed from the same mRNA it is, depending on the expression system, sorted either to the SP or to the mitochondria. This argues against an impact of codon choice on this alternative mechanism of protein sorting. Altogether, this leaves the impact of synonymous codons on translation velocity and folding of nascent proteins as the most likely explanation.

Analyses of Kesv sorting at the single cell level reveal that targeting of the channel not only depends on a combination of primary amino acid sequence and codon choice but also on the individual conditions of the cell that is expressing the protein. Depending on this cellular condition, a cell can interpret the same genetic information as a signal for sorting the channel protein either into the mitochondria, the secretory pathway or into both compartments; in other cell conditions the protein is degraded. In this context it is interesting to note that five different mammalian cell lines have distinctly different efficiencies for sorting the Kesv_wt_ channel to the mitochondria. This phenomenon is in good agreement with the finding that non-uniform distributions of rare/frequent codons in genes can generate patterns of local translation elongation/folding rates in an organism-specific manner [55]. Our experiments show that the state of the cell cycle is one cellular factor, which contributes to the sorting destiny of the Kesv channel in a given cell type.

A central dogma of molecular biology is that synonymous mutations have no effect on the primary amino acid sequence and hence on function of the resulting protein. These claims have been challenged by recent data underpinning important roles of codon choice in a wide range of events such as speed of protein synthesis, folding, stability and even function [7,8,9,10,11,12,13,14,15,16]. The present data now extend this scope of codon effects to the mechanism of protein sorting in mammalian cells. The apparent sensitivity of intracellular targeting of small channel proteins on codon bias suggests that the codon structure of a gene can together with a cell-state-dependent decoding mechanism which serves as a code for intracellular proteins trafficking. The data support a mechanism in which clusters of rare and common codons can serve together with the primary amino acid sequence as a signal for sorting of nascent membrane proteins to such fundamentally different destinations as mitochondria and secretory pathway. An intriguing consequence from the present study with model proteins is that the same sorting system may also operate with native membrane proteins. Any physiological or pathophysiological condition, like synonymous mutations [4] or variable concentrations of tRNA, [49,50] but also codon optimization in gene therapy [4], could direct the same protein to different membrane destinations in a cell.

## Figures and Tables

**Figure 1 cells-10-01128-f001:**
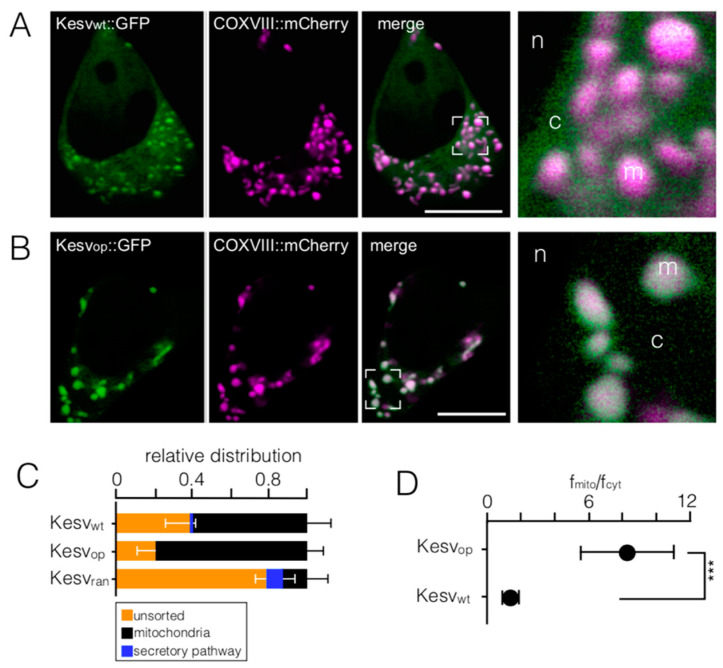
Codon optimization affects the sorting pattern of the Kesv channel. (**A**,**B**) Fluorescent images of HEK293 cells transfected with either Kesv_wt_ (**A**) or Kesv_op_ (**B**). Images show: GFP tagged channels (green, first column), fluorescence from mitochondrial marker (COXVIII::mCherry) (magenta, second column) and merging of magenta and green channels (third column). A magnification from areas marked in overlay images is shown in the fourth column. Letters in the images refer to cytosol (c), nucleus (n) and mitochondria (m). (**C**) Mean relative distribution (±SD; *n* ≥ 3) for localization of channels in mitochondria (black), secretory pathway (blue) or unsorted (orange) in HEK293 cells transfected with Kesv_wt_ (*n* = 259 cells), Kesv_op_ (*n* = 245 cells) or Kesv_ran_ (*n* = 120 cells). (**D**) Mean ratio ± SD of fluorescence intensity in mitochondria versus adjacent cytosol (f_mito_/f_cyt_) in cells transfected with Kesv_wt_ or Kesv_op_. A Student t-test predicts high statistical significance between the two conditions (*** *p* < 0.0001). Scale bars 10 µm.

**Figure 2 cells-10-01128-f002:**
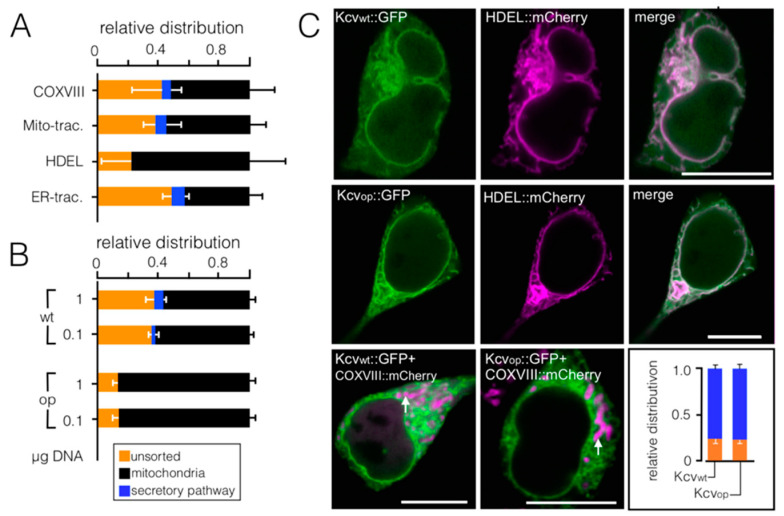
Codon-sensitive sorting is channel specific and not an artifact of the experimental system. (**A**) Mean relative distribution ± SD for localization of the Kesv channel in mitochondria (black), SP (blue), or unsorted (orange) in HEK293 cells transfected with Kesv_wt_. Images as in Figure 1 were analyzed by co-expression of the channel with marker proteins COXVIII::mCherry (*n* = 4, *n* = 63 cells) or HDEL::mCherry (*n* = 3, *n* = 55 cells) for mitochondria and ER, respectively. In separate experiments mitochondria and ER of Kesv_wt_ transfected cells were labeled with fluorescent dyes Mito- (*n* = 6, with ≥ 144 cells per condition) or ER-tracker, respectively (*n* = 3, ≥116 cells per condition). (**B**) Mean relative distribution of Kesv_wt_ and Kesv_op_ in HEK293 as in (**C**) from cells transfected transiently with either 0.1 or 1 µg DNA (*n* = 3 with ≥ 121 cells per treatment). (**C**) Fluorescent images of HEK293 cells transfected with Kcv_wt_ or Kcv_op_. Images in two top rows show: GFP tagged Kcv channels (green, first column), fluorescence from ER marker HDEL::mCherry (magenta, first and second row) and overlay of magenta and green channels in third column. Images in the third row show overlay of the GFP (green) and COXVIII::mCherry (magenta) channel for HEK293 cells transfected with either Kcv_wt_ or Kcv_op_. *Inset:* Mean relative distribution (n ≥ 220 cells) for localization of the channel in SP (blue), or unsorted channels (orange) in HEK293 cells transfected with Kcv_wt_ or Kcv_op_. Scale bars 10 µm.

**Figure 3 cells-10-01128-f003:**
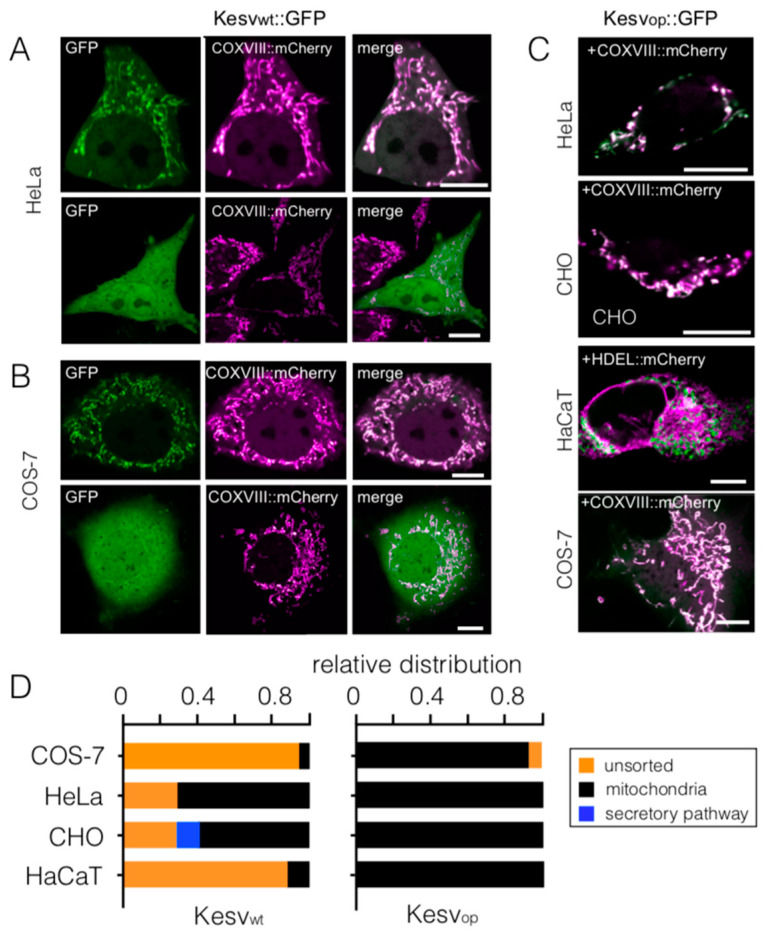
Sensitivity of channel sorting to codon optimization is conserved in mammalian cells. Fluorescent images of HeLa (**A**) and COS-7 cells (**B**) transfected with Kesv_wt_. In both cell types the channel exhibited either a clear-cut sorting to the mitochondria (top row) or a unsorted phenotype with GFP fluorescence throughout the cell (lower row). Images show: the GFP tagged Kesv channel (green, first column) and fluorescence from mitochondrial marker COXVIII::mCherry (magenta, second column). Merged images are in the third column. (**C**) Fluorescent images of different mammalian cells transfected with Kesv_op_. The images are overlays of GFP fluorescence (green) and fluorescence from mitochondrial marker COXVIII::mCherry (magenta) (HeLa, CHO, COS-7) or from ER marker HDEL::mCherry (HaCaT). Scale bars 10 µm. (**D**) Mean relative distribution ± SD for localization of the wt channel (Kesv_wt_, left) or codon-optimized channel (Kesv_op_, right) in mitochondria (black), secretory pathway (blue) or unsorted channels (orange). (Data from *n* ≥3; *n* ≥ 56 cells per condition).

**Figure 4 cells-10-01128-f004:**
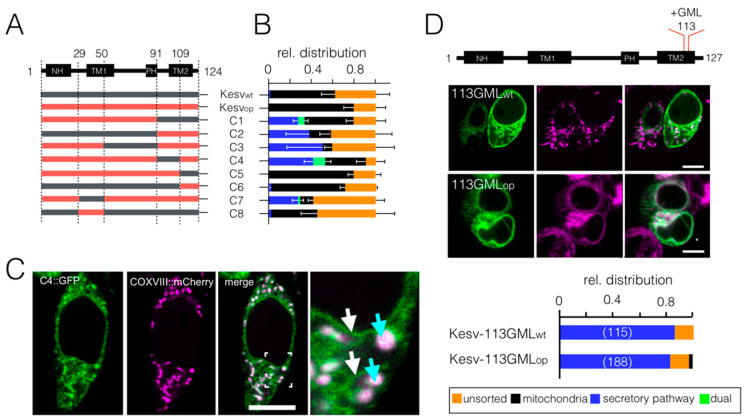
Complex sorting of the Kesv channel and its mutant in HEK293 cells transfected with chimera of genes with wt and optimized codons. (**A**) Schematic domain architecture of the Kesv channel monomer with transmembrane α-helixes including the N-terminal helix (NH), outer (TM1) and inner (TM2) transmembrane domain and pore helix (PH) (central panel) (top) and composition of Chimeras C1 to C8 comprising parts of the Kesv_wt_ (grey) and Kesv_op_ genes (red). (**B**) Mean relative distribution (±SD; *n* = 3, *n* ≥ 120 cells per chimera) of channels in mitochondria (black), SP (blue) and unsorted channels (orange) in HEK293 cells transfected with corresponding genes. The green bars represent cells in which the channel was present within the same cell in the mitochondria and in the SP. (**C**) Fluorescent image of a HEK293 cell transfected with Chimera C4. The images show distribution of GFP tagged chimera (green, left column), mitochondrial marker COXVIII::mCherry (magenta, second column), and an overlay of magenta and green channel (third column). The part indicated in the overlay is magnified in the fourth column with blue arrows and white arrows indicating presence of GFP in SP (white arrow) and mitochondria (blue arrow), respectively. (**D**) Top: schematic domain architecture as in A indicating position 113 in which the AA motive GML was inserted in TM2. Central: fluorescent images of HEK293 cells transfected with Kesv113GML from wt (113GML_wt_) or codon-optimized (113GML_op_) gene. Images show: the GFP tagged channel (green, first column) and fluorescence from mitochondrial marker COXVIII::mCherry (magenta, top) or from ER marker, HDEL::mCherry (magenta, down) as well as overlay of magenta and green channels (third column). Bottom: relative distribution of channels in mitochondria (black), SP (blue) and unsorted channels (orange) in HEK293 cells (numbers in brackets) transfected with corresponding genes. Scale bar in (**C**,**D**) 10 µm.

**Figure 5 cells-10-01128-f005:**
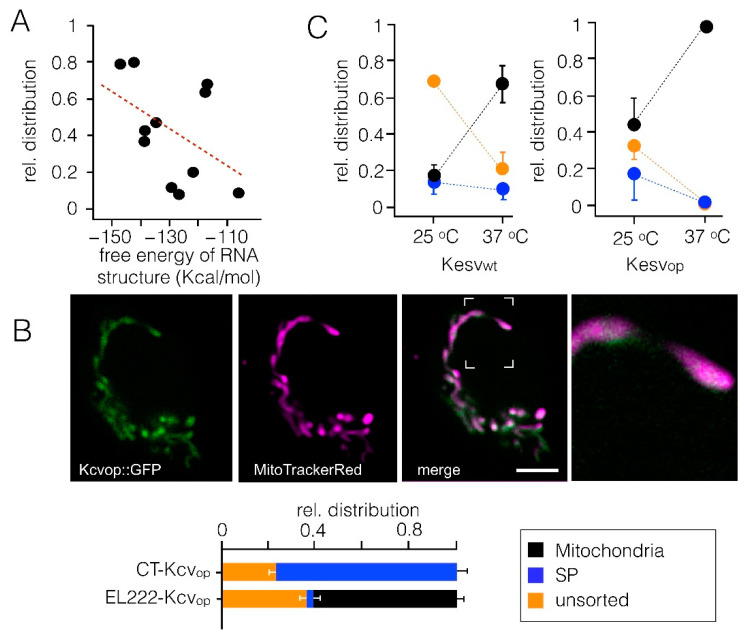
Sorting pattern of the Kesv channel as a function of parameters, which can affect protein synthesis. (**A**) Relative distribution of the Kesv channel in mitochondria from data in 4A as a function of estimated free energy of RNA structures derived for Kesv_wt_, Kesv_ran_, Kesv_op_ and chimeras C1-C8. Energies were calculated for channel coding the RNA sequence only. The line shows linear fit with a correlation coefficient of 0.44. (**B**) Light triggered transcription of Kcv_op_ channel shifts sorting propensity from SP to mitochondria. Top: confocal images of representative HEK293 cell expressing GFPtagged Kcv_op_ (green, 1st panel) and stained with MitoTrackerRed (magenta, 2nd panel); merging of green and magenta channels is shown in the 3rd panel with a blow up of the indicated area in the 4th panel. Bottom: relative distribution for localization of the Kcv_op_ channel in mitochondria, SP, or unsorted channels in HEK293. Protein was expressed in HEK293 cells by conventional transfection (CT) or under control of a light-sensitive EL222 system [25,44]. In the latter case, transcription was induced by a pulsed blue light of 120 µE, which was applied for 16 h prior to imaging. Light pulses were 10 s/20 s on followed by 60s of darkness. Data are from n cells in N independent experiments: CT-Kcv_Op_ *n* = 3, *n* = 237 and EL222-Kcv_Op_ *n* = 3, *n* = 118. Scale bars 10 µm. (**C**) Sorting of the channel to the three destinations (mitochondria: black, SP: blue and unsorted: orange) was estimated as in Figure 1 in HEK293 cells transfected with Kesv_wt_ (left panel) and Kesv_op_ (right panel). Cells were kept either at 37 °C or 25 °C. Lowering the temperature is unfavorable for channel sorting to the mitochondria. The lower temperature favors non-sorting and sorting to the SP. Mean values ± S.D. of *n* = 3 experiments with ≥ 270 cells per temperature. Color coding is the same as in B.

**Figure 6 cells-10-01128-f006:**
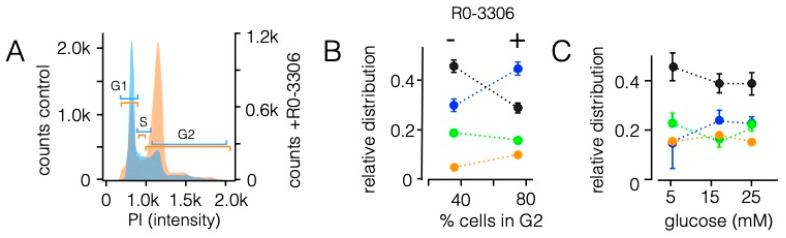
Sorting pattern of the Kesv channel in HEK293 cells is affected by cell cycle. (**A**) Analysis of DNA content in HEK293 cells by flow-cytometry in the absence and presence of cell cycle blocker R0-3306. Representative histograms of control cells (blue) and cells pretreated for 48 h with 7 μM RO-3306 (red) as function of Propidium Iodide (PI) intensity. The respective cell cycle phases are indicated by colored bars. (**B**) Relative distribution (±SD, *n* = 3, *n* ≥ 150 cells) of Chimera C1 in mitochondria (black), SP (blue), dual location in mitochondria and SP (green) as well as non-sorted channels (orange) as a function of cells in G2 in control cells (36 ± 3%) and RO-3306 treated cells (74 ± 1.5%). Cells were transfected 24 h after exposure to inhibitor and imaged 24 h later. (**C**) Mean relative distribution (±SD, *n* = 3, *n* ≥ 120 cells) of Kesv channel in mitochondria (black), SP (blue), dual location in mitochondria and SP (green) as well as non-sorted channels (orange) in HEK293 cells transfected with Chimera C1. Cells were supplied in culture medium with the indicated concentrations of glucose.

## Data Availability

All constructs used in this study are available on request.

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
