# Peer review of "Codon Bias Can Determine Sorting of a Potassium Channel Protein"

_cells, 2021, doi:10.3390/cells10051128_

Round 1

Reviewer 1 Report

It is very interesting and important paper on potassium channels sorting and codon role on this process. 
Data are presented in very clear way and the reviewer has only minor comments/questions/suggestions:
-    Localization of GFP tagged potassium channels were performed with the use of HEK293 cells. But also with CHO, HeLa, HaCaT and COS-7 cells. What was rational for choosing these cell lines (if any)?
-    It was not clear for reviewer in what experiments illumination setup (described lines 148-153) was used. Please provide info in Materials and Methods.
-    Because of K+ channels usage in this paper it may be better to use “potassium” instead of “ion” in title of the manuscript.
-    As ER tracker BODIPY-glibenclamide was used: please provide some rational for usage of this probe. Glibenclamide has various intracellular targets?
-    Energy status was modified (Figure  6c) by addition of various glucose concentration (lines 531-535). Would be possible to mention for example cellular ATP concentration (content) in this conditions?
-    Why presence of GFP in nucleus is evidence for miss-sorting?
-    One comment on language usage: word “bias” is used very much frequently (in various context?) in this paper. Line 2, 30, 81, 84, 296 “Codon bias”; Line 13, 51 “biased frequency”; Line 58 “unbiased insight”; line 190 “bias of manual classification”; line 232 “enhanced bias”; line 320 “codon biased sorting”; line 332 “bias for sorting” etc. It is not surprising but reviewer would suggest to limit number of its usage (to change to :”choice” or “usage” ?) in order to make manuscript even more clear. 

Author Response

Reviewer 1:

Localization of GFP tagged potassium channels were performed with the use of HEK293 cells. But also with CHO, HeLa, HaCaT and COS-7 cells. What was rational for choosing these cell lines (if any)?

Our response:

We have spelled out the motivation for using different cell lines (page 10, line 321-327). A detailed analysis of the interesting observation on why the sorting of the Kesv wt protein is so different in the different cell lines, is beyond the scope of this paper and will be addressed in a subsequent study.

Reviewer 1:

It was not clear for reviewer in what experiments illumination setup (described lines 148-153) was used. Please provide info in Materials and Methods.

Our response:

We have specified this in Materials and Methods (page 4, line 144):

“For illumination of cells in which the channel protein was expressed under control of the light sensitive EL222 system, a custom made …”

Reviewer 1:

Because of K+ channels usage in this paper it may be better to use “potassium” instead of “ion” in title of the manuscript.

Our response:

We follow the reviewer’s suggestion

Reviewer 1:

As ER tracker BODIPY-glibenclamide was used: please provide some rational for usage of this probe. Glibenclamide has various intracellular targets?

Our response:

The reviewer is correct that glibenclamide has many intracellular targets. But with a specific protocol the ER tracker Bodipy-glibenclamide can be used as a fluorescent marker for ER. The problem of unspecific labelling is not only relevant for the fluorescent ER tracker but for all stains of intracellular organelles. Also, the frequently used mito-trackers are staining other intracellular membranes. But with the appropriate protocols (e.g. concentration, staining time) the other membranes are not visible with respect to the strong fluorescent signal from the mitochondria. We have added in the materials section that we followed the manufacturers protocols.

Apart from these considerations we dod not have to worry too much in the present work about unspecific staining by the ER- and mito-trackers. We also used protein based ER and mitochondria markers (HDEL::mCherry, COXVIII::mCherry), which is very specific for the respective organelles. The results with dyes and protein markers are similar (Fig. 2a) suggesting that we did not overinterpret unspecific staining from fluorescent dyes.

Reviewer 1:

Energy status was modified (Figure  6c) by addition of various glucose concentration (lines 531-535). Would be possible to mention for example cellular ATP concentration (content) in this conditions?

Our response

Long term glucose starvation has many effects on cells not only on ATP concentration. We have extended our motivation and the background information on experiments with low and high glucose concentration in the medium (page 16, lines 533-539).  Since our treatment did not result in any appreciable effect on protein sorting, we did not perform additional experiments for specifying the effects of glucose starvation.

Reviewer 1:

Why presence of GFP in nucleus is evidence for miss-sorting?

Our response:

We have specified our claim (page 6, lines 228-230) on why the presence of GFP in the nucleus indicates cleavage from the hydrophobic channel. To avoid confusion in terminology we now avoid the term mis-sorting. We only speak of degradation, because that is the information, we get from the GFP in the nucleus.

Reviewer 1:

One comment on language usage: word “bias” is used very much frequently (in various context?) in this paper. Line 2, 30, 81, 84, 296 “Codon bias”; Line 13, 51 “biased frequency”; Line 58 “unbiased insight”; line 190 “bias of manual classification”; line 232 “enhanced bias”; line 320 “codon biased sorting”; line 332 “bias for sorting” etc. It is not surprising but reviewer would suggest to limit number of its usage (to change to :”choice” or “usage” ?) in order to make manuscript even more clear. 

Our response

We see the point of the reviewer. We have changed bias into synonymous word where possible throughout the manuscript.

Reviewer 2 Report

The manuscript presented by Engel et al. focuses on codon usage bias. The authors used K+ channels Kesv to show that the optimized codon changed the subcellular localization of the channel. The authors used an optimized codon of the channel fused with the GFP fusion and show that the fusion protein is targeted in the mitochondria. The authors performed several expriments to check whether subcellular markers (HDEL, mitochondria), expression level, temperature, cell cycle, cell types, etc. are responsible for the change of location. Among all the results, it appears that there are many factors that could affect the location of the canal as expected. The authors point out that the codon change is one possible explanation. The authors made different gene chimera and observed that their localization was different from that of the wild type or the optimized gene. This point is very interesting. However, many parameters seem to be involved. Despite this, the results were very interesting; there is no clear mechanism that could explain the optimized localization of the codon gene. Only observations and confirmation of known misallocation issues (eg cell type) are displayed. For the results presented in this manuscript, I have some concerns:

1) The localization of the labeled protein (Kesv-GFP) is crucial in all results. The authors assessed the localization based on microscopic observations. There is no automatic method to quantify and determine location accurately. Two different people had performed the classification of the protein sorting. I would recommend doing the experiment blind so as not to have bias from manual classification by the experimenter who knows what the expected results are.

2) Protein fusion is known to modify protein localization. The authors only did the C-terminal fusion. I would recommend doing the N-terminus fusion to check if the location is similar or changed.

Minor comment: References cited in the manuscript should appear as "[number]" and not "([number], authors et al. XXXX)"

Author Response

Reviewer 2:

The localization of the labeled protein (Kesv-GFP) is crucial in all results. The authors assessed the localization based on microscopic observations. There is no automatic method to quantify and determine location accurately. Two different people had performed the classification of the protein sorting. I would recommend doing the experiment blind so as not to have bias from manual classification by the experimenter who knows what the expected results are.

Our response

We appreciate the concern about the robustness and reproducibility of the microscopic analysis. All attempts to develop an algorithm for automated analysis unfortunately failed. However, the microscopic analysis by different individuals was very robust. We have added this information in the Materials and Methods section (page 5, lines 189-193). We had already performed the blinded experiments suggested by the reviewer in which we found that the results between three individuals deviated by less than 4%. This information was added in the material and method section.

Reviewer 2:

Protein fusion is known to modify protein localization. The authors only did the C-terminal fusion. I would recommend doing the N-terminus fusion to check if the location is similar or changed.

Our response

We share the concern of the reviewer in that the position of the tag could alter protein sorting. For the following reasons we however decided from the beginning of our study to keep the tag exclusively on the c-terminus:

  • The sorting of most membrane proteins (with the exception of tail anchored proteins) is decided on the N-terminus. In particular the beginning of the first transmembrane domain, which arrives in the nascent state of protein synthesis from the ribosome, is essential for the sorting destiny. If we would attach the GFP tag to the N-terminus we would create in terms of sorting, a completely different protein. For the sake of the complexity that we are already facing, we need to keep the experimental system as simple as possible.

  • The impact of the GFP on sorting would be essential in the context of studying the trafficking of a physiologically relevant cellular protein. However, in our study the protein is entirely artificial and orthogonal to the expressing cells. We can even view the entire channel plus GFP tag and linker as a model protein for studying the impact of codon usage.

Reviewer 2:

Minor comment: References cited in the manuscript should appear as "[number]" and not "([number], authors et al. XXXX)"

Our response

We corrected this mistake

Reviewer 3 Report

The authors of the manuscript entitled « Codon bias can determine sorting of a potassium channel protein » are eminent specialists of potassium channels, and more specifically potassium channels from viral origin. Two of these latter channels, Kcv and Kesv, cloned in two different phycodnaviridae species and sharing a strong structural homology, display nevertheless a completely different cellular sorting when expressed in mammalian cells. In their manuscript, the authors now show that codon optimization of the transcripts encoding Kcv and Kesv produces specific effects for both proteins. While cellular sorting of Kcv is not affected by codon optimization, Kesv is sorted to either mitochondria or the secretory pathway in a codon and cell cycle dependent-manner.   

I found this study to be scientifically sound and well conducted. Undoubtedly, this study will be of interest for a broad audience of readers. Nonetheless, I feel there are two points that should be clarified.

The first point relates to the results obtained with light-induced transcription. Those results, illustrated on the Fig.5B are quite impressive. I understand that the transcript produced either by conventional transfection (CT-Kcvop) or induced by light (EL222-Kcvop) will be the same, and that, according the authors’ reasoning (Line 470), similar mRNA structure should not affect protein sorting. But in both cases the translated protein (Kcvop::GFP) will also be the same. How explain different sorting of the same transcript and the same protein illustrated in Fig.5B? Did the authors realized similar experiments with Kcvwt which displays the same cellular sorting than Kcvop? or with Kesv-113GMLwt?

The second point relates to the amount of proteins produced by transient transfection. Basically, these amounts are no evaluated, and we don’t have any idea about differences between the amount of proteins produced with Kcv and Kesv. Is there any possibility that different amount of proteins may affect cell sorting independently of any signaling? The authors illustrate nuclear staining obtained with Kesvwt (Supplementary Fig.3B), and interpret this staining by the partial degradation of the tagged channel (Line 235). But it is not clear to me whether similar observations are done with the other channels (Kesvop, Kcv etc), and how this could influence authors’ conclusions. In the same way, the results obtained by varying the amount of plasmids used for transfection (Fig.2B) should be taken with more caution. There is not any correlation between the amount of plasmids used in transfection and the amount of proteins produced in individual cells and we don’t have any idea on what happens when this amount was divided by 10.

Minor points :

Line 147: “expresed under control”.

Line 155: “und 1% Penicillin”.

Line 216: “by analyzing the PI”, please define what PI is?

Line 241: “enhanced tendeny”.

Line 279-280: “had no appreciable impact on sorting 279 of Kesvwt and Kesvop (Fig. 2B) again suggesting that enhanced sorting”, I do not understand why it is suggested again?

Line 285: “KCVCV1”, KCVCV1 is not defined.

Line 372: “a stretch of codons >30 codons from the start”, the stretch begins at codon 29 on the Fig.4A.

Line 401: “the amino acid triplet GML”, it is not explained why this triplet has been chosen? It does not seem to be described in earlier studies [20] or [22].

Line 462: “-420kcal/mol”, this value does not fit with those on Fig5A.

Line 499: “cordon sensitive”

Figure 2: “(B)” and “(C)” were inverted in the caption.

Figure 6: X-axis legends are not appropriate. Fig6A: “% cells in G2” and “1.0k 1.5k”. Fig6B: “% cells in G2” for RO-3306 treated/untreated cells I guess. Fig6C: “glucose/mM” for glucose (mM) I guess.

Author Response

Dear Reviewer, thank you for the constructive comments and the close reading of the manuscript including spotting of mistakes. We have changed the manuscript along your suggestions including addition of an extra Figure in the supplement. We have we have clarified all open questions.

Reviewer: The first point relates to the results obtained with light-induced transcription. Those results, illustrated on the Fig.5B are quite impressive. I understand that the transcript produced either by conventional transfection (CT-Kcvop) or induced by light (EL222-Kcvop) will be the same, and that, according the authors’ reasoning (Line 470), similar mRNA structure should not affect protein sorting. But in both cases the translated protein (Kcvop::GFP) will also be the same. How explain different sorting of the same transcript and the same protein illustrated in Fig.5B? Did the authors realized similar experiments with Kcvwt which displays the same cellular sorting than Kcvop? or with Kesv-113GMLwt?

Our response: We have expanded the text, which explains the rational of this experiment (Line 473-478). The point, which is important here, is that the RNA is in both cases the same. This rules out the possibility that sorting is affected by a pre-sorting of different RNAs in the cells. The data imply that the difference in sorting is determined by a kinetic component. We have included this notion, but since we have no direct data, which show that the light controlled system (or codon optimization) is really speeding up translation, we refrain from speculating too much. We have also included a sentence which states that we are not able to explain the mechanism, which is underlying the difference in sorting in the optogenetic platform (line 487-488). But for the point of the present manuscript this is also not important.

Reviewer: The second point relates to the amount of proteins produced by transient transfection. Basically, these amounts are no evaluated, and we don’t have any idea about differences between the amount of proteins produced with Kcv and Kesv. Is there any possibility that different amount of proteins may affect cell sorting independently of any signaling? The authors illustrate nuclear staining obtained with Kesvwt (Supplementary Fig.3B), and interpret this staining by the partial degradation of the tagged channel (Line 235). But it is not clear to me whether similar observations are done with the other channels (Kesvop, Kcv etc), and how this could influence authors’ conclusions. In the same way, the results obtained by varying the amount of plasmids used for transfection (Fig.2B) should be taken with more caution. There is not any correlation between the amount of plasmids used in transfection and the amount of proteins produced in individual cells and we don’t have any idea on what happens when this amount was divided by 10.

Our response: We had been thinking about this point before and we have now added additional data in Fig. S5, which suggest that differences in protein expression are not at the root of different sorting destinies. We have therefore been analyzing the integrated fluorescence intensity of GFP in neighboring cells with Kesv either in mitochondria or SP. The data show no systematic difference which would indicate that sorting to mitochondria or SP is determined by high or low protein concentration. The text to this Fig. is found in lines 510-527.

Minor points :

Reviewer: Line 147: “expresed under control”.

Our response: Type corrected

Reviewer: Line 155: “und 1% Penicillin”.

Our response: Type corrected

Reviewer: Line 216: “by analyzing the PI”, please define what PI is?

Our response: Has been defined: Propidium Iodide (PI) in line 213 and in legend of Fig. 6

Reviewer: Line 241: “enhanced tendeny”.

Our response: Type corrected

Reviewer: Line 279-280: “had no appreciable impact on sorting 279 of Kesvwt and Kesvop (Fig. 2B) again suggesting that enhanced sorting”, I do not understand why it is suggested again?

Our response: It's an additional experiment: in one case Kesv is competing with a marker protein in the other case we looked at wt versus optimized channel. To avoid confusion we have rephrased the sentence.

Reviewer: Line 285: “KCVCV1”, KCVCV1 is not defined.

Our response: KcvPBCV1 has been defined

Reviewer: Line 372: “a stretch of codons >30 codons from the start”, the stretch begins at codon 29 on the Fig.4A.  

Our response: We corrected this as ≥30 codons

Reviewer: Line 401: “the amino acid triplet GML”, it is not explained why this triplet has been chosen? It does not seem to be described in earlier studies [20] or [22].

Our response: We added the information that the motive was discovered in a screening in which we inserted random triplets of amino acids. It would be out of scope of this paper to report the entire study. Text is added in lines 402-404.

Reviewer: Line 462: “-420kcal/mol”, this value does not fit with those on Fig5A.typo

Our response: Thanks for pointing this error out. We have corrected text and numbers.

Reviewer: Line 499: “cordon sensitive”

Our response: Type corrected

Reviewer: Figure 2: “(B)” and “(C)” were inverted in the caption.

Our response: Thanks for pointing out the error. It has been corrected

Reviewer: Figure 6: X-axis legends are not appropriate.

Fig6A: “% cells in G2” and “1.0k 1.5k”.

Our response: Thanks for pointing this error out; there was probably too much copy past. The mistake has been corrected.

Reviewer: Fig6B:  % cells in G2” for RO-3306 treated/untreated cells I guess.

Our response: In this case the labeling is correct. In the absence of the blocker the portion of cells in G2 is low (36±3) and increased to 74±1.5 in its presence. This is also explained in the legend. To help understanding the figure we added +/- R0-3306 on top of the bars.

Reviewer: Fig6C: “glucose/mM” for glucose (mM) I guess.

Our response: The labeling glucose/mM is not wrong. This type of axis labeling is frequently used in physics. But to avoid confusions we have changes it as suggested by the reviewer.

Round 2

Reviewer 2 Report

Engel et al. have improved the manuscript following the recommendations. Details about experimental procedures and some sentences were changed. However, my concerns on the N-terminal fusion has been not tried. I am not convincing by the answer. Any fusion could affect protein localization and it is important to try both side of the fusion. N-terminal fusion won’t create a different protein as the authors said because translation begins with GFP. I would suggest to fuse GFP at the N-terminal to show similar localization or not. This point is important because GFP is not supposed to target the channel. I highly recommend to do this control.

Author Response

Reviewer:

Engel et al. have improved the manuscript following the recommendations. Details about experimental procedures and some sentences were changed. However, my concerns on the N-terminal fusion has been not tried. I am not convincing by the answer. Any fusion could affect protein localization and it is important to try both side of the fusion. N-terminal fusion won’t create a different protein as the authors said because translation begins with GFP. I would suggest to fuse GFP at the N-terminal to show similar localization or not. This point is important because GFP is not supposed to target the channel. I highly recommend to do this control.

Our response:

We respectfully disagree with the reviewer. It has been shown in unbiassed assays that GFP can affect subcellular sorting of proteins including membrane proteins when attached to the N-terminus but not to the C-terminus in the proteins of interest [1]. This makes perfectly sense since the complex machinery of protein sorting is “reading” from N- to C-terminus. By the time the C-terminally attached GFP is translated the decision of sorting has already been made. In the case of N-terminal tagging, the GFP is already folded when the protein of interest, e.g. the channel, emerges from the ribosome exit tunnel. In this situation the GFP may mask the relevant sorting signals in the emerging polypeptide. This is particularly relevant in short membrane proteins like the ion channels in the present study. In these types of proteins, the very first amino acids, which generate the leading hydrophobic polypeptide chain in the nascent protein, contain relevant sorting information. Hence the decision has already been made by the time translation reaches the ion channel protein [2]. For these reasons we do not want to include experiments on the sorting of N-terminally tagged channels. To clarify our reasoning, we include the following paragraph plus relevant references in the materials and methods section: 

“Because sorting of small membrane proteins is very sensitive to the structure of the nascent N-terminal polypeptide chain, which emerges from the ribosome [14] and since an N-terminal GFP tag is more likely to corrupt protein sorting than a C-terminal tag [24] all channel variants were exclusively tagged on the C-terminus via a linker to eGFP and inserted in a peGFP-N2 vector.”

Saying this, we fully agree with the reviewer, in that N-terminally tagged proteins will teach us something more about sorting of the channels of interest. But the information will be on a different level of complexity than what we studied in this paper, namely the impact of codon bias. In fact in the case of N-terminal tagging, we would also need to address the question on whether the codon usage of the GFP affects sorting. This would add even one more parameter.

[1] Palmer E, Freeman T. Investigation into the use of C- and N-terminal GFP fusion proteins for subcellular localization studies using reverse transfection microarrays. Comp Func Genom 20045, 342-353. 

[2] Pechmann S, Frydman J. Evolutionary conservation of codon optimality reveals hidden signatures of cotranslational folding. Nat. Struct. Mol. Biol201320, 237–243.